# Acoustic Features of the Impact of Laser Pulses on Metal-Ceramic Carbide Alloy Surface

**DOI:** 10.3390/s24165160

**Published:** 2024-08-10

**Authors:** Sergey N. Grigoriev, Mikhail P. Kozochkin, Artur N. Porvatov, Evgeniy A. Ostrikov, Enver S. Mustafaev, Vladimir D. Gurin, Anna A. Okunkova

**Affiliations:** Department of High-Efficiency Processing Technologies, Moscow State University of Technology STANKIN, Vadkovskiy per. 3A, 127994 Moscow, Russia; s.grigoriev@stankin.ru (S.N.G.); porvatov_artur@mail.ru (A.N.P.); e.ostrikov@stankin.ru (E.A.O.); e.mustafaev@stankin.ru (E.S.M.); v.gurin@stankin.ru (V.D.G.); a.okunkova@stankin.ru (A.A.O.)

**Keywords:** acoustic emission, concentrated energy flows, laser, power density, pulses, self-oscillation, spectral analyses

## Abstract

Technologies associated with using concentrated energy flows are increasingly used in industry due to the need to manufacture products made of hard alloys and other difficult-to-process materials. This work is devoted to expanding knowledge about the processes accompanying the impact of laser pulses on material surfaces. The features of these processes are reflected in the acoustic emission signals, the parameters of which were used as a tool for understanding the accompanying phenomena. The influence of plasma formations above the material surface on self-oscillatory phenomena and the self-regulation process that affects pulse productivity were examined. The stability of plasma formation over time, its influence on the pulse performance, and changes in the heat flux power density were considered. Experimental data show the change in the power density transmitted by laser pulses to the surface when the focal plane is shifted. Experiments on the impact of laser pulses of different powers and durations on the surface of a hard alloy showed a relationship between the amplitude of acoustic emission and the pulse performance. This work shows the data content of acoustic emission signals and the possibility of expanding the research of concentrated energy flow technologies.

## 1. Introduction

The use of concentrated energy flows of high power density in material processing is a rapidly growing trend in modern industry. The uniqueness of these results lies in their ability to process super-hard materials, act on minimal volumes of matter, convert them from a solid to a plasma state in a split second, interact with parts of minimal rigidity, and maintain environmental friendliness during operations [1]. These technologies, including laser processing, electrical discharge machining, electron-beam machining, and others, have different energy carriers and application areas but are all based on a concentrated flow of thermal energy affecting a small spatial segment. The advantage of localizing the substance’s volume and short exposure period to these technologies also presents significant difficulties in studying the accompanying processes’ kinetics. The analysis of acoustic emission (AE) signals accompanying the effect of concentrated energy flows on the surface of a material has recently been increasingly used as a tool for understanding such processes such as electrical discharge machining, laser and electron-beam machining [2,3,4], and laser and plasma deposition methods [5,6].

There are several types of monitoring systems that can be used for electrophysical processes, divided by the nature of the data being monitored: electrical, optical, and acoustic [7,8]. Control and parameter regulation processing in modern CNC machines for electrophysical machining is based on electrical pulses since there is often no visual access to the processing zone [9]. At the same time, optical monitoring tools are commonly used for those technologies where there is optical access to the processing zone, and it is possible to monitor the molten pool and the amount of material ejected from the processing zone visually [10,11]. There is no visual access to the treatment area during laser pulse processing or the formation of wells with a depth-to-diameter ratio greater than 0.5. Moreover, during the formation, the well is covered with a plasma cloud of the sublimated material, which at some point begins to interfere with further processing, and the machining itself can be classified as a self-regulating, self-oscillating process [12]. Control of electrical parameters during laser pulse processing is not as obvious as during electrical discharge machining. Acoustic emission monitoring represents an alternative that allows the evaluation of the performance of laser pulses and offers prospects for creating a multi-parameter equipment control system [13].

The monitoring of vibroacoustic signals has long been used in flaw detection as a non-destructive testing method [14,15,16,17] and as a control for other machining methods [18,19,20]. The acoustic emission in the form of elastic waves propagating through the part’s material and its fastening points on the equipment arises from a thermal shock created by an energy pulse. As a result of the rapid heating of a small volume of a substance, its expansion occurs, generating elastic and plastic deformations in the surrounding space. Deformation waves propagate through an elastic medium over long distances at the speed of sound in solids [21,22,23,24], allowing them to be recorded using vibration sensors at a distance from the process. The processes of rapid thermal expansion and a number of accompanying phenomena contribute to the energy of vibroacoustic signals. For example, phenomena that accompany the development of defects in the form of cracks and dislocations [25,26], which arise during the restructuring of the crystal lattice [27,28,29] during the crystallization and melting of a substance [30,31,32,33].

The purpose of this work is to study the relationships between the parameters of acoustic emission accompanying the action of laser pulses on the surface of various materials with the output processing characteristics, which include productivity and quality of the resulting surface, and assess the possibility of monitoring the performance of laser processing by the parameters of acoustic emission signals based on this knowledge. Experimental studies were carried out to obtain information about the capabilities of the parameters of acoustic emission signals to display physical phenomena that arise during the action of laser pulses on the surface of the material being processed, affecting the processing performance and quality indicators of the resulting surface.

## 2. Materials and Methods

### 2.1. Workpiece Materials

Cutting inserts made of VK6 (M05 by ISO) and T15K6 (P10 by ISO) hard alloys were used in the experiments [34,35,36,37]. The chemical compositions of the alloys are provided in Table 1. In the experiments with the laser pulse performance and the relation of acoustic emission amplitude parameters with the position of the focal plane (Δ), 3 mm thick plates made of D16 aluminum alloy (AlCu4Mg1, AISI 2024) and R6M5 high-speed steel (AISI M2) were used. The chemical compositions of those materials are provided in Table 2 and Table 3.

### 2.2. Laser Processing Equipment and Vibroacoustic Monitoring Means

The experiments were carried out on a laser machine for precision processing, SharpMark Fiber GT 60 (LLC Sharplase, Moscow, Russia). Figure 1 shows the machine’s working area with the installed cutting insert and accelerometers. The technical characteristics of the laser unit are presented in Table 4. The main characteristics of the accelerometer, model AP2037-100 (LLC GlobalTest, Sarov, Nizhny Novgorod region, Russia), are provided in Table 5. Signals from the accelerometers were recorded on a PC using the installed software developed by MSTU Stankin (Moscow, Russia) for further analysis. As seen in Table 5, the linear frequency range for measuring vibration acceleration signals extends only to 15 kHz, but it is much broader than the operating range. The operating range is a linear range where the accelerometer conversion coefficient is constant. For this range, the accelerometer’s passport indicates a conversion factor that allows us to obtain vibration measurement results in natural units (m/s^2^). However, the frequency response of the accelerometer is much broader than the linear range, extending into the region of low and high frequencies. It is also possible to evaluate vibrations there, but only in conventional units or mV. Measurements in a series of experiments are conducted in the same frequency range to compare them with each other to see how many times the results of different measurements differ in the frequency range beyond the linear range. It can be noted that the largest conversion coefficient is in the resonance region, which makes it possible to observe vibrations of small amplitude.

### 2.3. Investigation of the Samples and Methods of Acoustic Emission Signal Parameters’ Presentation

#### 2.3.1. Influence of Plasma Plume Formations on the Parameters of Acoustic Emission, Self-Oscillations, and Self-Regulating Processes

For the research of the influence of plasma plume formations on the parameters of acoustic emission and the mechanisms of self-oscillations and self-regulating processes in laser processing, an experiment was conducted in which 1000 pulses with a power of 60 W, a duration of 500 ns, and a frequency of 10 kHz were applied at one point on the workpiece surface made of the T15K6 hard alloy. The interval of the pulses was 0.1 ms.

#### 2.3.2. Effect of a Sequence of Laser Pulses on the Volume of Removed Material (Q)

For the research on the effect of a sequence of laser pulses on the volume of removed material (Q), five experiment conditions were carried out with a 3 mm thick aluminum alloy plate. The different numbers of pulses (5, 10, 20, 40, 80 pulses) with a frequency of 10 kHz, a power of 60 W, and a duration of 500 ns were addressed to the end face of the plate to produce each well. For each type of well, diameter (D) and depth (L) were measured, and the volume of the removed material (Q) was calculated. A thin plate was scanned on an industrial X-ray tomograph FILIN CT-1500 [38] (TESTRON LLC, St. Petersburg, Russia) to determine the wells’ profile and the ejecta volume. The profile of one well for each experiment was shaded for better visibility. The original shadow images of the wells on the tomograph screen can also be seen. The volume of the wells was estimated from the assumption that the sections perpendicular to the well axis are circular and the entire well’s image is a cavity. The images were divided into sections that were approached by cylindrical or conical surfaces.

#### 2.3.3. Relation of the Parameters of Acoustic Emission (K_f_ Parameter) and Power Density

The surface of plates made of R6M5 high-speed steel and D16 aluminum alloy was used to research the parameters of acoustic emission signals obtained in laser processing with a change in power density. The power density of the laser pulse can be changed by changing the following:The position of the focal plane (changing the distance from the lens to the surface under processing);The power of the generator.

For the research on the influence of the position of the focal plane, the plate made of R6M5 high-speed steel was installed at an angle relative to the focal plane. The lines of wells were produced at an area set from 3 mm higher to 3 mm lower than the focal plane. The inclination should noticeably change the power density at the lens’s focus depth in the range of ±1 mm. For the convenience of acoustic emission amplitude presentation, the K_f_ parameter was used as the ratio of acoustic emission amplitudes at low and high frequencies (A_1_ and A_2_):(1)$$K_{f}=\frac{A_{1}}{A_{2}}.$$
It should be noted that minimal K_f_ corresponds to the removal caused mainly by intense material evaporation [4]. Optical microscopy was carried out on an Olympus BX51M instrument (Ryf AG, Grenchen, Switzerland).

For the research on the influence of the generator’s power, the plate made of D16 aluminum alloy was processed with exposure laser pulses of 500 ns duration, a frequency of 2 kHz, and a power of 60 W.

#### 2.3.4. Influence of the Processing Mode (Laser Power N and Duration τ) on Well and Ejecta Volume (Q and V)

The applied value of acoustic emission parameters is more significant if it is established that those parameters are closely related to the characteristics of the processing performance and processed surface quality. For this purpose, an experiment was carried out on the impact of laser pulses with different powers and durations on the surface of cutting inserts made of T15K6 hard alloy. The conditions and results of the experiment conducted on 15 processing modes are presented in Table 6. The experiment consisted of making a line of 10 wells using those modes at a frequency of 2 kHz. Five passes were made along these wells at those modes (Figure 2). The profilograms of the wells were obtained using a Dektak XT stylus profilometry (BRUKER company, Billerica, Massachusetts, USA) [39]. The data on well and ejecta volumes were obtained using the Dektak XT profilometer software (Vision 64 software on a Windows 7 OS, version 1).

Mathematical processing of the received data is needed to present information about the relationship between the parameters of acoustic emission signals and the characteristics of wells produced on cutting inserts made of the T15K6 hard alloy. The experimental planning technique enabled mathematical models and graphs showing the relationship between the acoustic emission parameters (A_1_ and A_2_) with laser power N and pulse duration τ.

Since the aim of this work is to assess the possibility of monitoring the performance of laser processing based on the parameters of acoustic emission signals, the relationship between performance Q and the power N and pulse duration τ were studied using the same methodology. It is important to compare the dependencies of acoustic parameters and performance characteristics, since the presence of a close connection may have practical significance for using acoustic parameters in monitoring means.

### 2.4. Features of the Self-Oscillatory and Self-Regulating Processes under the Impact of Concentrated Energy Flows on the Surface

During conventional blade processing, the cutting is accompanied by flows of chips that can have many different shapes. Not all of those chip shapes are conducive to the smooth operation of technological equipment. First of all, in relation to equipment operating in automatic mode: there are no chips in the usual sense when processing with concentrated energy flows, but it does not exclude problems associated with removing destructible material. In electrical discharge machining, the processing productivity and quality of the processed surfaces largely depend on the stability of the discharge gap state. Instead of chips, erosion products enter the discharge gap during processing, which are particles of the removed metal. This debris appears under discharge pulses: first, the surface material turns into metal vapors, and then the particles enter the working fluid and solidify [40,41]. Erosion products absorb part of the discharge energy in the gap and reduce the power density of concentrated energy flows affecting the surface under processing [3]. They are removed from the gap using forced flushing. However, this is not always completely possible. Then, the concentration of erosion products increases in the gap, reducing the share of useful work (output of energy).

A decrease in processing productivity reduces the concentration of erosion products, restoring the share of useful work (fully or partially). Negative feedback arises that carries out self-regulation of the process [32]. This correlation’s properties are unstable due to the random nature of the conditions for removing particles and the size of the particles of erosion products. Large particles are more difficult to remove. Despite the effects of self-regulation, it leads to the accumulation of erosion products and a decrease in the power density supplied to the workpiece surface, which leads to a decrease in the evaporation/sublimation of the material. These phenomena lead to the localization of discharges and short circuits with wire breakage [42,43,44].

Since monitoring acoustic emission signals allows us to observe changes in the performance of the pulses in the discharge gap, the effects of self-regulation of the electrical discharge machining can also be seen in the recordings of acoustic emission signals. Figure 3 shows an example of the changes in the amplitude of acoustic emission in the processing area of the VK6 hard alloy from the beginning until the wire electrode breaks.

In works [1,2,3,4,5,6], it is shown that a decrease in the amplitude of acoustic emission signals in the high-frequency range relative to the low-frequency range indicates a decrease in the proportion of the evaporated substance relative to the volume of the melt in processing by concentrated energy flows. Figure 3 shows that the amplitude of oscillations in the high-frequency range gradually decreases relative to the amplitude in the low-frequency range, where it even increases slightly. This suggests that the result of the opposite processes of accumulation and removal of erosion products favors increasing their concentration. If you look at the acoustic emission signal envelope (Figure 3a), you can see that the change in the performance of the electrical discharge machining is uneven. Numerous spikes and drops in the amplitude of the acoustic emission signal are visible. These fluctuations can be attributed to the self-regulating process of electrical discharge machining. Such changes have all the signs of a self-oscillating process since the electrical discharge machining system itself selects the moments of decrease and increase in the concentration of erosion products depending on a set of random factors, which are determined only by the process of material destruction itself and the conditions for removing erosion products. There is no strict periodicity characteristic of self-oscillations as in many other mechanical systems with elastic elements, where the frequency of self-oscillations is of usually close to the natural frequencies the dynamic system [45]. In the case of electrical discharge machining, random factors do not allow for obtaining a self-oscillation frequency close to a specific value.

Since technologies using concentrated energy flows rely on thermal energy flow with high power density, the accompanying phenomena should be mutually similar [4]. In this case, even during laser processing, self-oscillations should be present, which are displayed in the parameters of the acoustic emission signals. Figure 4 shows a schematic representation of the effect of a laser pulse on a metal surface. The pulsating nature of the increase in the well’s depth during laser melting was noted earlier [46]. However, tracing this phenomenon’s reflection in the acoustic emission signal parameters is interesting. The essence of the appearance of self-oscillations during laser melting is similar to the processes described for electrical discharge machining.

As shown in Figure 4, when a certain temperature is exceeded, a cloud of material in a vapor state and a plasma plume are formed above the well. The density of the plasma increases with increasing temperature and absorbs an increasing amount of laser radiation. Reducing the energy supplied to the surface during processing reduces the amount of evaporated substance, reducing plasma density in the plume. It increases the energy supply to the surface and increases the amount of matter evaporation and the plasma density in the plume. These processes determine the self-oscillatory mode. In electrical discharge machining, the erosion products in the gap act as a regulator. In laser processing, it is a vapor–plasma plume above the well.

## 3. Results

### 3.1. The Influence of Plasma Formations on the Self-Oscillations and Self-Regulating Processes under the Impact of Concentrated Energy Flows on the Surface

Figure 5 shows a reflection of the self-oscillations and the self-regulating process in the parameters of acoustic emission signals when 1000 pulses of power of 60 W, a duration of 500 ns, and a frequency of 10 kHz were applied at one point on the workpiece surface made of the T15K6 hard alloy. The figure shows RMS amplitude envelopes of acoustic emission signals in two frequency ranges: 5–17 kHz and 35–50 kHz. Thus, the RMS envelope corresponded to addressing pulses to one point with an interval of 0.1 ms. It can be seen that the amplitudes in both ranges begin to decrease after 2 ms. This suggests that the plasma density has become high enough and has begun to absorb part of the laser radiation energy. However, the decrease does not occur evenly but with local maxima, followed by a further decline. The amplitude of the acoustic emission signals decreased by more than an order of magnitude on the 10th ms.

The inset of Figure 5 shows the change in RMS amplitude in the range of 35–50 kHz after the 10th ms. It can be seen that after the amplitude decreased to a low level, its further decline stopped, but its modulation acquired an almost periodic nature. The process has stabilized, and the depth of modulation of the acoustic emission amplitude and its frequency have become almost constant. However, practically no useful work is performed in such a self-oscillating mode. That is evident from the rapid decrease in the acoustic emission amplitude, showing that almost all the energy is absorbed by the plasma plume formed above the well.

The rate at which the amplitude of the acoustic emission signal decreases (Figure 5) indicates a high rate of formation of the cloud of vapor and plasma. The cloud’s dissipation rate—in other words, the reverse process—can be estimated by the behavior of the acoustic emission amplitude as the interval between the pulses increases.

Figure 6 shows a recording of acoustic emission signals in the 5–17 kHz range when laser pulses are sequentially applied to 24 points on 50 passes. At a pulse frequency of 2 kHz, pulses were addressed to each well every 12 ms. In this case, the acoustic emission amplitude on the first pass was 8.8 times greater than on the last. At the 50th pulse, the acoustic emission amplitude dropped by 12.9 times. With a ratio of intervals between pulses of 1:120 (Figure 5 and Figure 6), the acoustic emission amplitude decreased only 1.5 times, indicating the stability of the formed plasma plume.

In Figure 6, the decrease in acoustic emission amplitude becomes noticeable after seven laser passes. Suppose we estimate the decrease in amplitude in 10 passes in Figure 5 and Figure 6, and then a more significant difference is noted:The amplitude decreases by 2.5 times with an interval of 0.1 ms;The amplitude decreases by 1.15 times at an interval of 12 ms.

Thus, an increase in the interval between pulses increases the dissipation of the vapor and plasma cloud over the well. However, this effect is relatively weak, and ventilation of the laser irradiation area is required.

If we compare the records in Figure 5 and Figure 6, there is no process modulation in Figure 6. The amplitudes of the acoustic emission signals decrease monotonically. This suggests that with a longer time interval between pulses, the dynamic component of the vapor and plasma cloud is partially dissipated and partially goes into a static state, increasing the density of the cloud. Thus, the amount of material removed decreases with each new pulse.

Figure 7 shows a parallel record of RMS amplitudes in two frequency ranges under the same conditions as in Figure 6. The records are brought closer to each other to show they are almost equidistant. That is, the ratio of the amplitude of component 1 to the amplitude of component 2 throughout the laser exposure process is almost constant.

As seen in Figure 7, as the number of passes increases, the amplitude of the acoustic emission signal components decreases. However, the ratio of amplitudes in the low- and high-frequency ranges remains close to a constant value.

In works [3,4,6], it was shown that changes in the ratio of amplitudes in the low- and high-frequency ranges when processing with the concentrated energy flows indicate changes in the power density of the thermal effect on the material surface. The amplitude in the high-frequency component of acoustic emission increases faster than in the low-frequency range with increasing power density. A decrease in power density has the opposite effect on the amplitude ratio [1,4]. In Figure 7, the acoustic emission amplitude decreases across all frequency ranges, but the amplitude ratio remains almost similar. Since the productivity of processing with concentrated energy flows is related to the acoustic emission amplitude by a monotonic dependence, a decrease in acoustic emission amplitudes indicates a decrease in productivity [42]. Nevertheless, maintaining the constant amplitude ratio indicates a constant power density. Based on these results, the following conclusion is drawn: forming a plasma cloud under laser irradiation reduces the area where the power density is high. In other words, the area of the real influence of the laser irradiation on the material is continuously narrowing, trending towards zero.

### 3.2. Acoustic Studies on the Influence of a Plasma Plume on the Performance of Laser Pulses

Figure 8 shows well profile images obtained on a tomograph for five experiment conditions with different numbers of pulses applied to produce a well on an aluminum alloy plate: numbers of pulses—5, 10, 20, 40, 80; a frequency of 10 kHz, a power of 60 W, and a duration of 500 ns. The number of pulses, well diameter (D), depth (L), and volume of the removed material (Q) are indicated in Table 7.

Figure 9 shows images of a series of wells after processing with 40 pulses. The wells’ profiles have complex shapes. However, there is no evidence that the material was removed from the entire volume. This is especially true for the lower part of the wells treated with 40 and 80 pulses. That increases the interest in acoustic emission signals if they can be used to indicate instant processing performance. It is especially important if there is no direct access to the processing zone.

The decrease in RMS amplitude of acoustic emission when a series of pulses is applied to one well is shown in Figure 5, Figure 6 and Figure 7. The decrease in RMS amplitudes is determined by the later pulses forming a smaller RMS amplitude (Figure 9b). As a result, the average acoustic emission energy throughout the entire process decreases with an increasing number of pulses. Accordingly, the RMS amplitude also decreases. However, a decrease in RMS amplitude can also occur due to a displacement of the focal plane relative to the material surface when the well deepens. In this case, the area of the laser spot on the surface increases with the same pulse power, and power density decreases.

Figure 10 shows changes in the parameters of acoustic emission signals obtained by laser processing the surface of a plate made of R6M5 high-speed steel, with the lens’s focus depth in the range of ±1 mm (the change in the position of the focal plane).

As can be seen in Figure 10, the ratio of acoustic emission amplitudes at low and high frequencies (K_f_ parameter) varied from 25 to 7. The material was not removed when the values of the K_f_ parameter were large (Figure 10b). At minimum K_f_, the removal was mainly due to intense material evaporation [4]. This should affect the well volume reduction relative to the volume of melt ejecta on the material surface. The dependencies shown in Figure 10 are natural: a displacement of the focal plane leads to an increase in the focal spot area and a natural decrease in power density distribution. As the power density decreases, the plasma formation and the evaporated material share decrease. A further decrease in power density reduces the amount of melt, and the process is limited only by surface heating [46,47,48].

Figure 11 shows this, using the example of exposure laser pulses of 500 ns duration, at a frequency of 2 kHz, and a power of 60 W on a surface of the plate made of D16 aluminum alloy.

The amplitude ratio (K_f_) decreases following a linear dependence with increasing power N of the supplied pulses (Figure 11a). The displacement of the focal plane above and below the processing surface (∆ = ±1 mm) causes an increase in the K_f_ parameter due to an increase in the laser spot area. In this case, a shift of 1 mm does not change the power density noticeably with a focal depth of ±0.7 mm. Thus, the K_f_ parameter also changes by 19%. Figure 10 and Figure 11 are violated when a sequence of identical pulses is supplied to one well and a material vapor and plasma plume is formed (Figure 4).

Figure 12 shows an analysis of the parameters of the acoustic emission signal that accompanied the experiment, shown in Figure 8 and Figure 9.

Figure 12 shows a combined image of the RMS amplitudes in the low- and high-frequency ranges of the acoustic emission signal that accompanied the laser processing. Each RMS amplitude peak reflects the averaged acoustic emission signal that accompanied the process of sequential exposure to single pulses on 15 wells (one laser pass). There were 80 passes in total, but only the first 37 passes are shown in the figures since the signal amplitudes become small and approach the interference level. It is noteworthy that if the K_f_ parameter was 2.7 on the first pass, then the amplitudes of the signals in both frequency ranges became almost equal by the 35th pass, and the K_f_ parameter fluctuated around ~1.0 until the end of all passes. In this experiment, a decrease in useful energy is accompanied by an increase in power density. Thus, the action of laser pulses on later passes is limited to the impact on small surface areas, where the phenomenon of evaporation of matter begins to predominate. Following the results of the analysis of acoustic emission signals, most of the energy of the laser pulses is absorbed with a large number of pulses supplied to one well. Furthermore, the energy that reaches the material surface generates shorter pulses with low amplitude, which are more characteristic of evaporation processes [3,4]. The results presented above speak of acoustic emission signals as an effective tool for understanding processes using concentrated energy flows.

### 3.3. Research into the Relationship between Acoustic Emission Parameters and the Performance of Laser Pulses with Different Powers and Pulse Durations

Figure 13 shows profilograms of the wells produced on the surface of a cutting insert made of T15K6 hard alloy. There are three areas where lines of wells were obtained in accordance with the modes presented in Table 6. Each area is characterized by constant pulse duration and different power values. Table 8 provides the RMS amplitude values for two frequency ranges and the well and ejecta volumes.

Figure 13b,c show that changes in the power of laser pulses naturally affect the well profiles and ejecta. It is visually noticeable that a decrease in pulse power leads to a decrease in the well profiles. Figure 13d,e show wells obtained in mode 11 on an enlarged scale. Figure 13e,f show sections of wells obtained in modes 11 and 14. It should be noted that there are no data on the volumes Q and V for modes 10 and 15 because there are no traces of processing on the surface of the cutting insert (Figure 13a–c).

## 4. Discussion

### 4.1. Relationship of the RMS Amplitude of the Acoustic Emission Signal in the Low (A_1_) and High (A_2_) Frequency Ranges with the Power (N) and Pulse Duration (τ)

Figure 14 shows the dependence A_1_(N, τ) in general form and as projections of individual sections onto coordinate planes.

Figure 14 shows that an increase in the power N and pulse duration τ leads to an increase in the amplitude of the acoustic emission signal in the relatively low-frequency range of 10–28 kHz. However, an increase in pulse power N causes a faster increase in amplitude than pulse duration τ. The nature of the presented dependences is such that the dependence of the RMS amplitude A_1_ on pulse duration τ or power N can be approximated by a linear function with a minimum error in the range of more than 30 W. Expression (1) is a mathematical description of the dependence A_1_ (τ, N):(2)$$A_{1}(\tau ,N)=539.0\cdot 10^{-4}N^{2}+9.64\cdot 10^{-4}\tau^{2}-532.1\cdot 10^{-2}N-947.1\cdot 10^{-3}\tau +100.6\cdot 10^{-4}N\cdot \tau +219.6.$$

The standard deviation of Model (1) was less than 3% of the maximum value. It should be noted that laser processing was not possible in modes with a laser pulse power of less than 30 W. Therefore, the increase in the A_1_ values at a power of 20 W in Figure 14a,c results from the model’s extrapolation beyond the experimental values.

Similar dependencies were plotted for a higher-frequency range of acoustic emission of 32–70 kHz (A_2_). Figure 15 shows a graphical presentation of the dependence A_2_ (τ, N). The dependencies shown in Figure 14 and Figure 15 are quantitatively different, but the trends in the RMS amplitude of acoustic emission repeat.

A comparison of Figure 14 and Figure 15 shows the coincidence of the main trends. This is especially evident at high powers and durations of laser pulses. An increase in the power N and pulse duration τ leads to an increase in the RMS amplitude A_2_, but an increase in power N has a more substantial effect.

### 4.2. Relationship between the Well Volume (Q) and the Power (N) and Pulse Duration (τ)

Figure 16 shows a graphical representation of the dependence Q(N, τ) in general form and as projections of sections onto coordinate planes.

In Figure 15, the graphs are similar to the graphs in Figure 14. Here, the dependence of well volume Q on power N is also more pronounced, as for the RMS amplitude in the low-frequency range A_1_ in Figure 14. An increase in the pulse duration is also accompanied by an increase in both well volume Q and RMS amplitude in the low-frequency range A_1_ but with a smaller gradient than the influence of pulse power N. The mathematical description of the dependence Q (N, τ) is represented by Expression (3):(3)$$Q\left(\tau ,N\right)=851.2\cdot 10^{-3}N^{2}+166.2\cdot 10^{-3}\tau^{2}-88.74N-141.3\tau +133.6\cdot 10^{-2}N\cdot \tau +172.85.$$
The standard deviation of Model (3) from the experimental values was less than 3% of the maximum value.

### 4.3. Comparison of Dependencies of RMS Amplitude in Low-Frequency Range A_1_ (N, τ) and Well Volume Q (N, τ)

The compared characteristics of the experimental values were first presented as percentages of their maximums for a parallel presentation of the dependences of RMS amplitude in low-frequency range A_1_ and well volume Q on power N and pulse duration τ. The conversion results are shown in Table 9, where index “p” indicates conversion to percent units. According to those data, the dependences A_1p_ (N, τ) and Q_p_ (N, τ) were plotted (Figure 17).

The surfaces shown in Figure 17 can be described as Expressions (2) and (3) by polynomials of the second degree in two variables. The standard deviation in this case will be 2.4% for A_1p_ and 2.1% for Q_p_. Nevertheless, since Figure 17 shows that the depicted surfaces at laser pulse powers above 30 W resemble flat surfaces, it is interesting to consider linearized models represented by a linear polynomial consisting of three terms represented by Expressions (4) and (5). In the designation of the amplitude and well volume, “L” indicates the linearization of the model.
(4)$$A_{1pL}\left(N,\tau \right)=1.96N+0.14\tau -107.07,$$
(5)$$Q_{pL}\left(N,\tau \right)=1.98N+0.15\tau -113.43.$$
The standard deviation of Models (4) and (5) from the experimental values became greater but remained at an acceptable level: 9% for A_1pL_ (N, τ) and 8.3% for Q_pL_ (N, τ). As can be seen from Expressions (4) and (5), the coefficients of the variables are close. Thus, the planes cannot formally be considered parallel, but such an assumption is possible when determining experimental values. The standard deviation between the planes described by Expressions (4) and (5) is 4.3%. The discrepancy observed in the region of low laser pulse powers is explained by the absence of processing traces and the influence of interference on the acoustic emission signal.

### 4.4. Relationship between the Volumes of Material Adhering to the Surface (V) Being Processed with Processing Modes (N, τ) and the Well Volume (Q)

Figure 18 shows an example of the distribution of material removed from the well and adhered to the surface of the cutting insert surrounding the well. This adhering material participates in the formation of the surface microrelief and determines the need for further processing. For a differentiated assessment of the well’s volume and the adhering material’s volume, the position of the cutting insert surface relative to the well and the adhering material was specified. This made it possible to determine the volume of the well below the cutting insert plane and the volume of material adhering to the surface. The volume of material adhering to the surface was calculated as a percentage of the volume of material removed from the well. The dependence V (N, τ) was plotted, where V was the percentage of the well volume Q. Figure 19 shows the dependence V (N, τ) in general form and as projections of sections onto coordinate planes.

The graphs in Figure 18 show that the relative volume of material adhering near the wells decreases with increasing power and pulse duration. As in the previous dependencies, changes in V are more noticeable when power N changes than when pulse duration τ changes. The mathematical expression describing the dependence V (N, τ) is represented by Expression (6):(6)$$V(N,\tau )=154.4\cdot 10^{-3}N^{2}-0.4358\cdot 10^{-3}\tau^{2}-26.04N-0.9441\tau +22.9\cdot 10^{-3}N\cdot \tau +926.7.$$
The standard deviation of Model (5) reaches 7%. This deviation is acceptable given the complex and largely random shape of the material ejected on the surface.

## 5. Conclusions

In the studies conducted, acoustic emission signals were used to understand the internal processes accompanying the action of laser pulses on the surface of hard alloys. This made it possible to observe several phenomena during concentrated energy flow processing and to establish relations between the acoustic emission parameters and the quality indicators of laser processing.

It was demonstrated using the analysis of acoustic emission signals that the self-regulating effects of concentrated energy flow impact processes arise due to the occurrence of erosion products during electrical discharge machining and plasma plumes during laser exposure on conductive surfaces. The formation of a plasma plume acts as a regulator in a self-regulating system with negative feedback. That leads to the self-oscillatory processes reflected in acoustic emission signals.

The study of acoustic emission signals made it possible to establish that the plasma plume is sufficiently stable over time. For this reason, reducing the frequency of laser pulses does not significantly reduce its ability to absorb the energy of laser pulses.

Experiments with sequential delivery of laser pulses into one well showed that the presence of a plasma formation almost does not reduce the power density of the supplied energy but quickly reduces the area of influence of concentrated energy flows on the workpiece surface, which reduces the productivity of new pulses almost to zero.

Experiments with varying the power and duration of laser pulses showed that the volumes of material removed are closely related to the RMS amplitude of the acoustic emission signal. This makes it possible to monitor the current performance of laser processing from the accompanying acoustic signal. It was also found that an increase in the power and duration of laser pulses leads to a decrease in the volume of material emissions near the well under processing relative to the well volume.

Acoustic emission signals in a wide frequency range can display phenomena that accompany the concentrated energy flow impact on materials. They can be used as a tool for understanding the difficult-to-observe phenomena of these technologies and for observing current processing performance and possible deviations of the specific processing conditions.

## Figures and Tables

**Figure 1 sensors-24-05160-f001:**
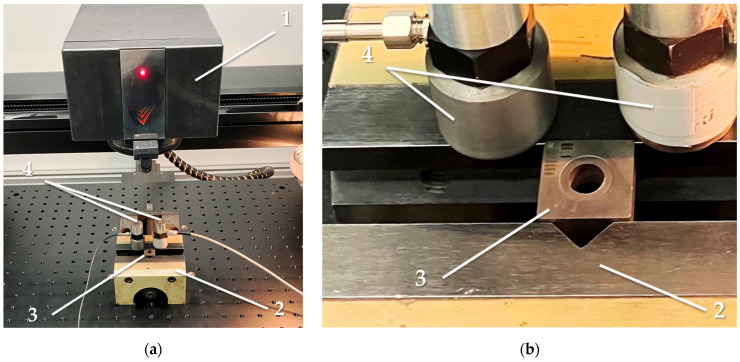
Laser processing machine, SharpMark Fiber GT 60: (**a**) general view of the working area; (**b**) cutting insert made of the hard alloy T15K6 in a vice and accelerometers, where (1) is a laser module, (2) is a vice, (3) is a fastened workpiece, and (4) shows accelerometers.

**Figure 2 sensors-24-05160-f002:**
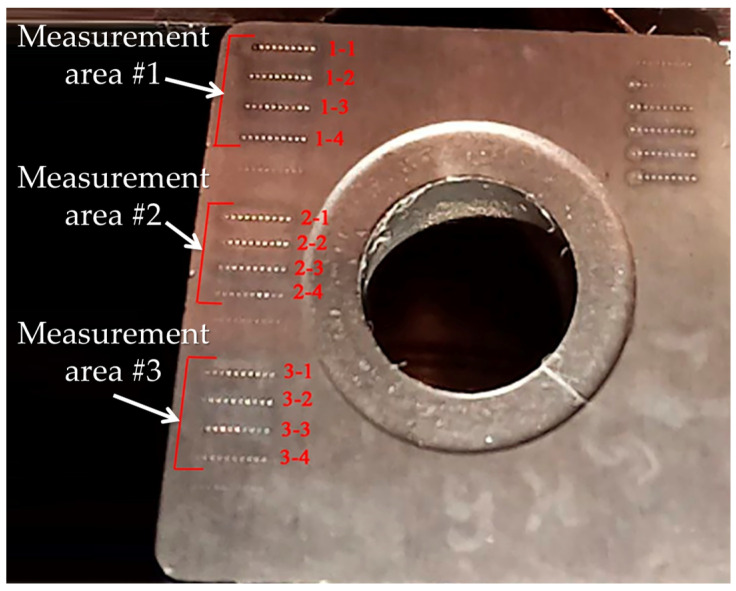
The general view of cutting insert made of T15K6 alloy.

**Figure 3 sensors-24-05160-f003:**
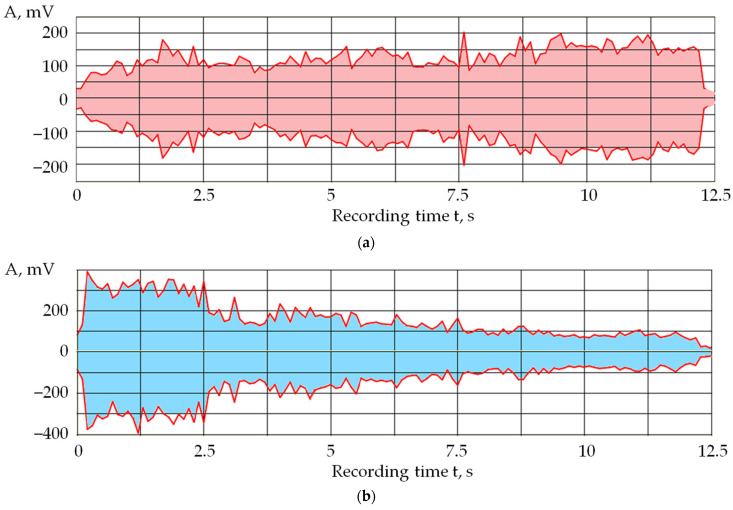
Envelopes of acoustic emission signals in two frequency ranges from the moment the electrical discharge machining begins until the wire electrode breaks: (**a**) low-frequency range; (**b**) high-frequency range.

**Figure 4 sensors-24-05160-f004:**
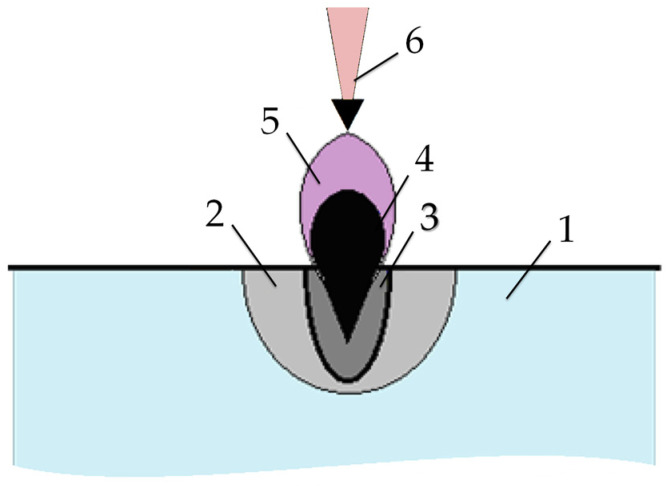
Formation of various regions when laser pulses act on the surface of a substance: (1) is a region of elastic deformations; (2) is a plastic deformation region; (3) is melt; (4) is vapors of matter; (5) is plasma; (6) is laser beam.

**Figure 5 sensors-24-05160-f005:**
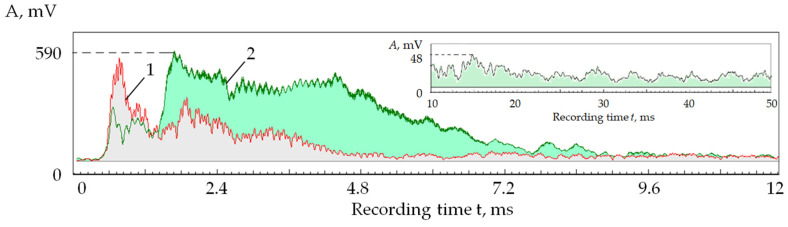
Interaction of laser pulses with the T15K6 alloy and its display in acoustic emission signals: RMS amplitude of the acoustic emission signal in the frequency ranges 5–17 kHz (1) and 35–50 kHz (2); inset shows the change in RMS amplitude in the range of 35–50 kHz after the 10th ms.

**Figure 6 sensors-24-05160-f006:**
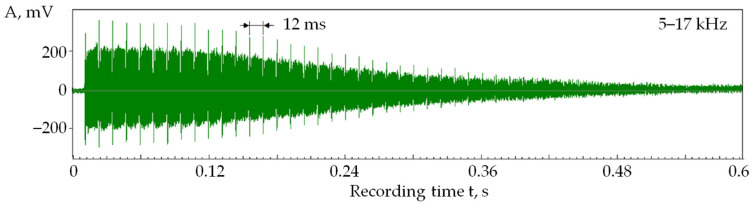
RMS amplitude of the acoustic emission signal in 5–17 kHz when sequentially applying laser pulses to 24 points in 50 passes (pulse frequency of 2 kHz ≈ interval of 12 ms).

**Figure 7 sensors-24-05160-f007:**
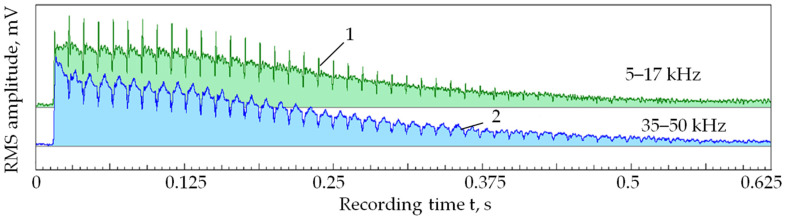
RMS amplitude of the acoustic emission signal in 5–17 kHz (1) and 35–50 kHz (2) when sequentially applying pulses to 24 points in 50 passes.

**Figure 8 sensors-24-05160-f008:**
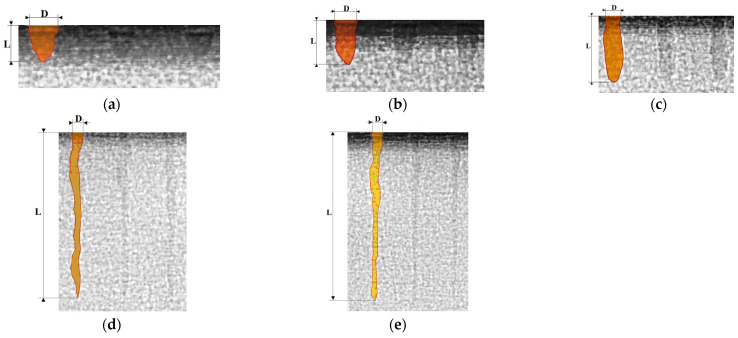
Well profiles obtained on a tomograph FILIN CT-1500 after exposure to different numbers of laser pulses: (**a**) 5 pulses; (**b**) 10 pulses; (**c**) 20 pulses; (**d**) 40 pulses; (**e**) 80 pulses.

**Figure 9 sensors-24-05160-f009:**
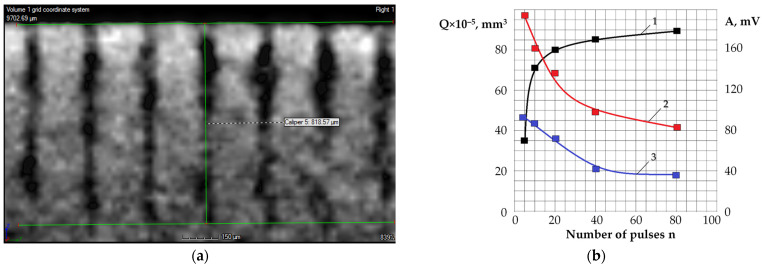
The wells on the tomograph (**a**) and the dependence of the volume of the wells and the RMS amplitudes of the acoustic emission signals during processing with different numbers of pulses (**b**): 1 is the volume of the wells Q with a different number of pulses; 2 is RMS amplitude of acoustic emission in the range of 5–17 kHz; 3 is RMS amplitude of acoustic emission in the range of 35–70 kHz.

**Figure 10 sensors-24-05160-f010:**
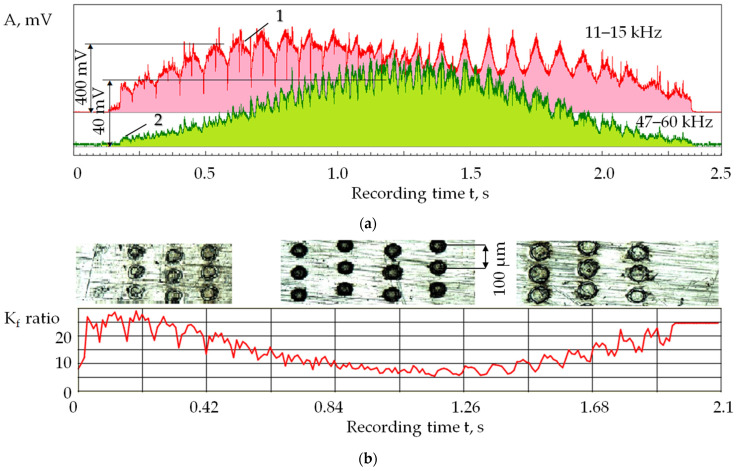
Acoustic emission signals when processing a plate made of R6M5 high-speed steel installed with an inclination relative to the focal plane: (**a**) RMS amplitudes in the ranges of 11–15 kHz (1) and 47–60 kHz (2); (**b**) the ratio of RMS amplitudes 1 and 2 (K_f_ parameter) when displacement of the focal plane is above and below the processing surface, ∆ = ±1 mm (pictures in Figure 10b show traces of laser pulses on different parts of the surface).

**Figure 11 sensors-24-05160-f011:**
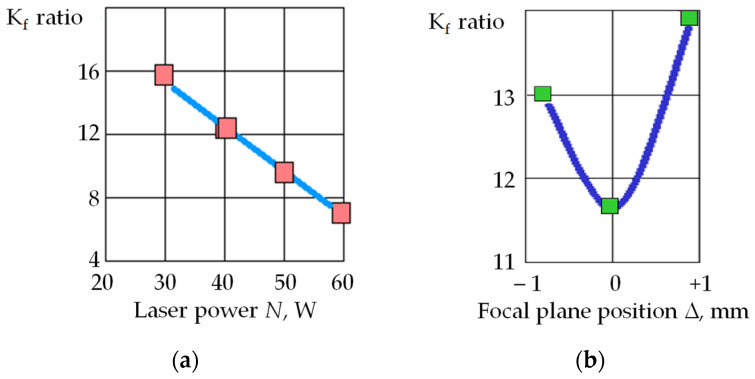
The ratio of RMS amplitudes (K_f_ parameter) of acoustic emission signals in the ranges of 3–23 kHz and 31–51 kHz in D16 aluminum alloy processing: (**a**) to the power of laser pulses N, W; (**b**) to the position of the focal plane ∆, mm.

**Figure 12 sensors-24-05160-f012:**
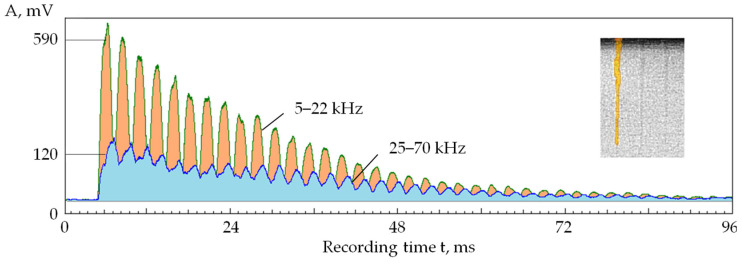
RMS amplitudes of the acoustic emission signal in two frequency ranges when producing a line of 15 holes with the sequential supply of pulses in 80 passes (the first microseconds of the recording).

**Figure 13 sensors-24-05160-f013:**
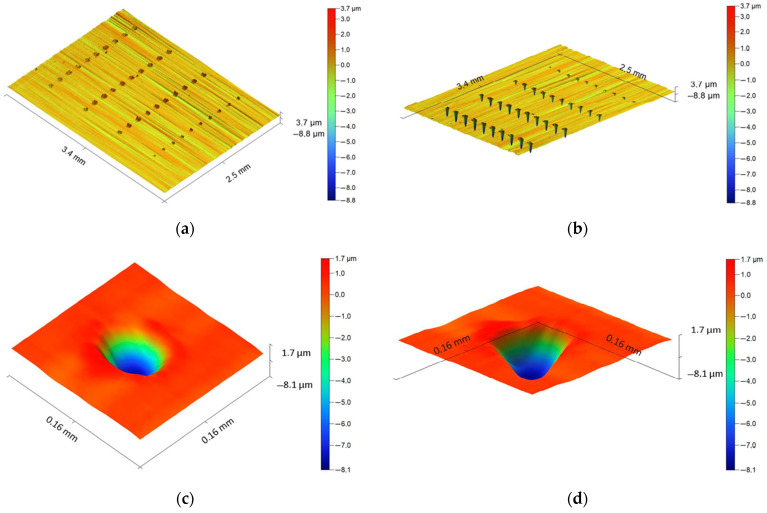
Well profiles on the surface of T15K6 cutting insert obtained using the Dektak XT stylus profilometer: (**a**) 3D profilogram (area 3, top view); (**b**) 3D profilogram (area 3, bottom view); (**c**) 3D profilogram (area 3-1, top view); (**d**) 3D profilogram of a well of (area 3-1, bottom view); (**e**) linear profilogram (mode 11); (**f**) linear profilogram (mode 14).

**Figure 14 sensors-24-05160-f014:**
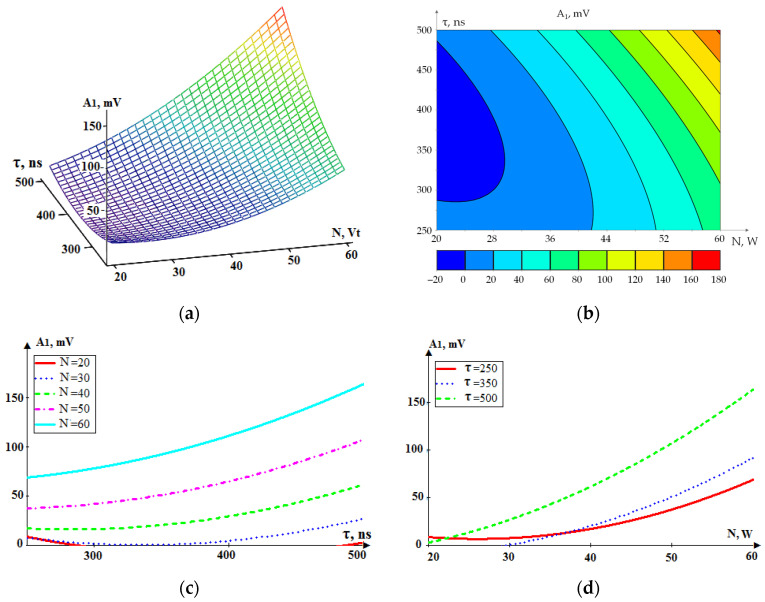
The dependence of RMS amplitude of the acoustic emission signal in the low-frequency range (10–28 kHz, A_1_) on the power (N) and pulse duration (τ): (**a**) general view; (**b**–**d**) projections of lines of equal level onto coordinate planes.

**Figure 15 sensors-24-05160-f015:**
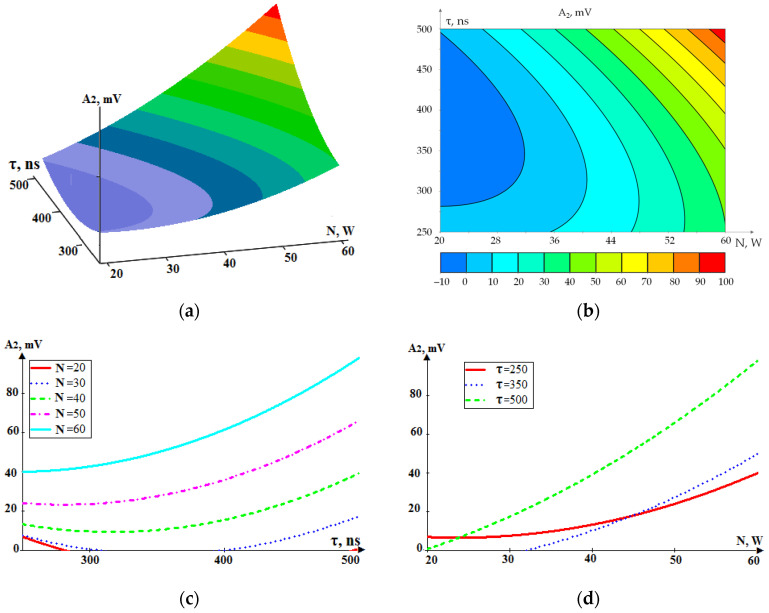
The dependence of RMS amplitude of the acoustic emission signal in the high-frequency range (32–70 kHz, A_2_) on the power (N) and pulse duration (τ): (**a**) general view; (**b**–**d**) projections of lines of equal level onto coordinate planes.

**Figure 16 sensors-24-05160-f016:**
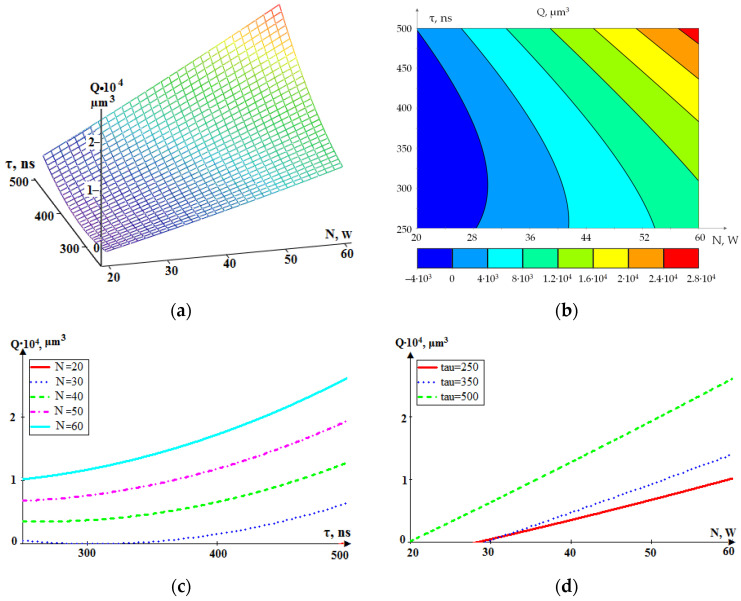
The dependence of the volume of removed material Q on the power (N) and pulse duration (τ): (**a**) general view; (**b**–**d**) projections of lines of equal level onto coordinate planes.

**Figure 17 sensors-24-05160-f017:**
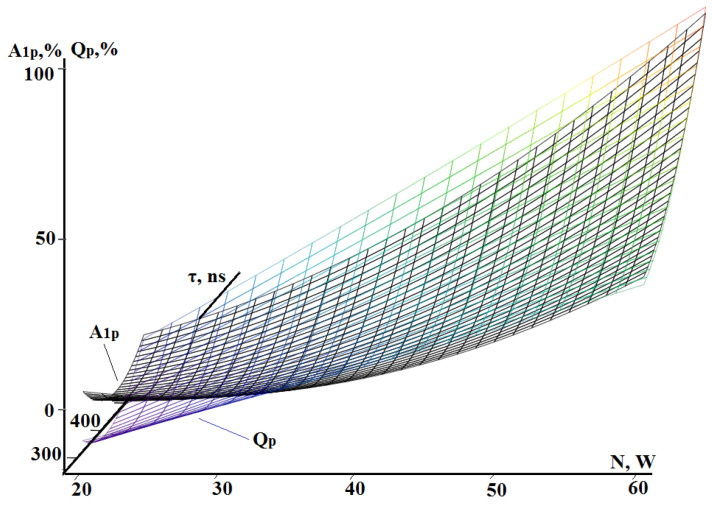
Joint graphical representation of dependencies of RMS amplitude in low-frequency range A_1_ (N, τ) and well volume Q (N, τ) converted to percent units (indexed “p”).

**Figure 18 sensors-24-05160-f018:**
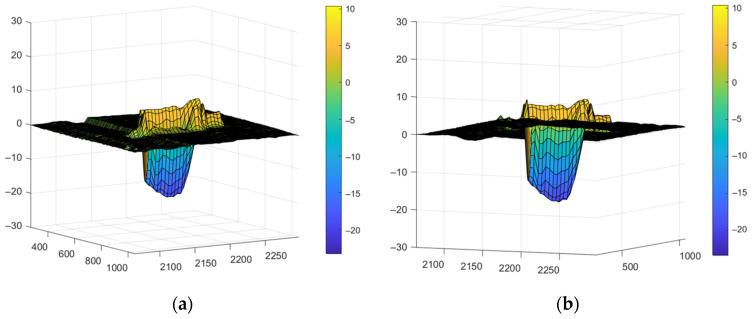
The distribution of material removed from the well and adhered to the surface of the cutting insert surrounding the well: (**a**) top view; (**b**) bottom view.

**Figure 19 sensors-24-05160-f019:**
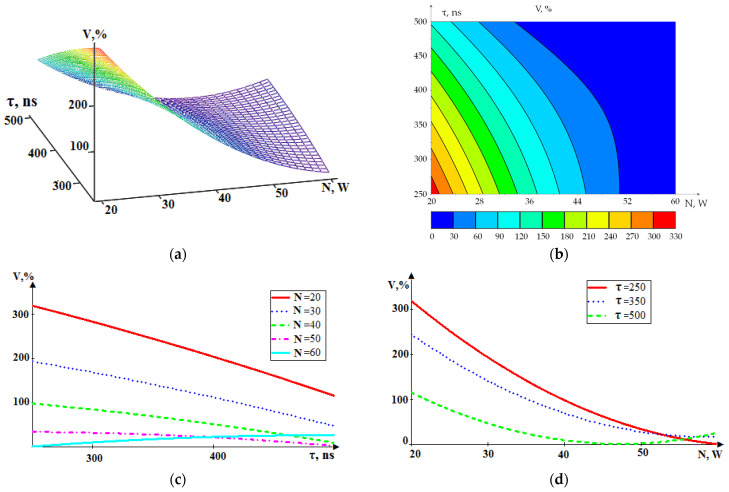
The dependence of the volume of material adhered to the surface surrounding the well, as a percentage of the volume of the well, depending on the power (N) and duration (τ) of laser pulses: (**a**) general view; (**b**–**d**) projections of lines of equal level onto coordinate planes.

**Table 1 sensors-24-05160-t001:** The chemical composition of M05 and P10 hard alloys, %.

Hard Alloy	Component, %			
	WC	TaC	TiC	Co
VK6 (M05)	92	2	-	6
T15K6 (P10)	79	-	15	6

**Table 4 sensors-24-05160-t004:** Technical characteristics of the laser unit.

Characteristic	Description
Laser type	Pulsed fiber laser
Operating wavelength, nm	1064
Maximum power, W	60
Energy in a radiation pulse, mJ	2
Pulse duration, ns	From 2 to 500
Software	SHARPLASE SOFT^TM^ based on C++ (version 190627)

**Table 5 sensors-24-05160-t005:** Technical characteristics of the accelerometer.

Characteristic	Measuring Unit	Value
Conversion factor	mV/m/s^2^	10
Linear frequency range	Hz	0.5–15,000
Resonant frequency in axial direction	kHz	>45
Noise level, RMS ^1^ (1 Hz–10 kHz)	m/s^2^	<0.0035

^1^ Root mean square.

**Table 6 sensors-24-05160-t006:** Processing modes of the experiment with cutting insert made of T15K6 alloy.

**Measurement Area**	**Experiment Number**	**Processing Mode**	**Pulse Duration τ, ns**	**Pulse Power N, W**
1	1–1	1	500	60
	1–2	2		50
	1–3	3		40
	1–4	4		30
	1–5	5		20
2	2–1	6	350	60
	2–2	7		50
	2–3	8		40
	2–4	9		30
	2–5	10		20
3	3–1	11	250	60
	3–2	12		50
	3–3	13		40
	3–4	14		30
	3–5	15		20

**Table 2 sensors-24-05160-t002:** The chemical composition of D16 aluminum alloy, %.

Element	Al	Cu	Mg	Mn	Fe	Si	Zn	Ni	Ti
%	90.8–94.7	3.8–4.9	1.2–1.8	0.3–0.9	Up to 0.5	Up to 0.5	Up to 0.3	Up to 0.1	Up to 0.1

**Table 3 sensors-24-05160-t003:** The chemical composition of R6M5 high-speed steel, %.

Element	Fe	W	Mo	Cr	V	C	Mn	Si	Ni	Co	Cu	P	S
%	From 78.4	5.5–6.5	4.8–5.3	3.8–4.4	1.7–2.1	0.8–0.9	0.2–0.5	0.2–0.5	Up to 0.6	Up to 0.5	Up to 0.25	Up to 0.03	Up to 0.025

**Table 7 sensors-24-05160-t007:** Parameters of the formed well profiles.

Number of Pulses	Diameter D, mm	Depth L, mm	Volume of Removed Material Q × 10^–5^, mm^3^
5	0.089	0.102	35.0
10	0.076	0.159	72.0
20	0.062	0.273	83.0
40	0.061	0.818	86.0
80	0.049	1.047	89.0

**Table 8 sensors-24-05160-t008:** Results of the experiment with cutting insert made of T15K6 alloy.

Processing Mode	Pulse Duration τ, ns	Pulse Power N, W	RMS Amplitude in 10–28 kHz A_1_, mV	RMS Amplitude in 32–70 kHz A_2_, mV	Well Volume Q, µm^3^	Ejecta Volume V, %
1	500	60	166.0	100.0	25,900.0	12.0
2		50	110.0	70.0	20,277.9	13.0
3		40	60.0	35.0	12,079.0	25.2
4		30	22.2	13.0	5966.0	45.7
5		20	4.0	3.0	291.6	105.6
6	350	60	84.7	42.4	12,999.0	22.9
7		50	49.7	26.2	9670.0	25.3
8		40	22.4	12.8	4857.0	50.2
9		30	7.7	4.14	781.8	157.5
10		20	0	0	0	-
11	250	60	72.4	42.0	10,819.0	7.3
12		50	40.1	27.5	6456.0	44.1
13		40	15.4	12.8	3258.0	62.8
14		30	3.0	2.3	346.0	211.6
15		20	0	0	0	-

**Table 9 sensors-24-05160-t009:** Function comparison data for RMS amplitude in low-frequency range A_1_ (N, τ) and well volume Q (N, τ) converted to percent units (p).

Functions	Processing Mode														
	1	2	3	4	5	6	7	8	9	10	11	12	13	14	15
RMS amplitude A_1p_, %	100	66	36	13	2.4	51	30	13.5	4.6	0	43.6	24	9.3	1.8	0
Well volume Q_p_, %	100	78.3	46.6	23	1.1	50.2	37.3	18.8	3	0	41.8	24.9	12.6	1.3	0

## Data Availability

The original contributions presented in the study are included in the article, further inquiries can be directed to the corresponding author.

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
