# Peer review of "Acoustic Features of the Impact of Laser Pulses on Metal-Ceramic Carbide Alloy Surface"

_sensors, 2024, doi:10.3390/s24165160_

Round 1
Reviewer 1 Report
Comments and Suggestions for Authors
This article is devoted to the issues of diagnostics of laser processing of materials. Its subject matter corresponds to the profile of journal Sensors and may be of interest to its readers. However, it is not entirely clear from the text how original the materials of the paper are? Judging by the bibliographic list, the authors have been working on the use of acoustic waves in laser processing of materials for a long time. Therefore, if any new scientific results were obtained recently, it should be very clearly commented on in the paper. By the way, the very fact of diligent self-citation (7 works!) in itself, it raises many questions, including from the point of view of scientific ethics.
Another remark concerns the fact that the introduction to the article is very brief, and there is no mention of other methods for diagnosing physical processes during the laser processing of materials. The authors should compare their proposed approach with other methods and justify its advantages.
Comments on the Quality of English LanguageModerate editing of English language required
Author Response
Response to Reviewer 1 Comments
Dear reviewer,
Thank you so much for your kind evaluation of our work. We agree with all your proposals and comments and have modified the manuscript accordingly.
We hope the manuscript will be suitable for publishing in Sensors and attract many potential journal readers with your comments. The introduced corrections in the text of the manuscript are marked yellow.
Kind regards,
Authors.
Reviewer comments
Point 1: This article is devoted to the issues of diagnostics of laser processing of materials. Its subject matter corresponds to the profile of journal Sensors and may be of interest to its readers. However, it is not entirely clear from the text how original the materials of the paper are? Judging by the bibliographic list, the authors have been working on the use of acoustic waves in laser processing of materials for a long time. Therefore, if any new scientific results were obtained recently, it should be very clearly commented on in the paper. By the way, the very fact of diligent self-citation (7 works!) in itself, it raises many questions, including from the point of view of scientific ethics.
Response 1: Thank you so much for your kind remark. The research group of our university indeed works decades in the field of research of acoustic emission in machining in application to different technologies. Our research group achieved many remarkable results from the scientific and practical point of view based on the idea that every process produces specific sound waves in an elastic body. That is confirmed by many publications in the most authoritative scientific journals and patents. However, not many other research groups have developed this idea for practical application. We have developed monitoring systems that can be used for cutting equipment, even working at high speeds, and control processing remotely; special software for the monitoring systems, criteria of preferable processing parameters with feedback to the CNC to adapt the speed based on the data of the acoustic emission in tool wear. Many important relationships between the process factors and product parameters were achieved during the work. Another complex of scientific work was devoted to electron beam technologies and electrical discharge machining. Multiparameter CNC systems were built based on the received data.
This article is devoted to an absolutely new topic that was never covered by other scientists. Firstly, the nature of the self-oscillation process during pulse laser processing (self-regulating plasma cloud formation that prevents further well formation) was shown. Secondly, it was shown that received acoustic emission data has a regular character and can be informative in a wide range of frequencies. It was shown that acoustic emission data is more informative in the low-frequency range of 10-32 kHz and reflects the data achieved for processing productivity. The obtained graphs can be approximated according to the known data of the known natural law. The necessary data can be added to the manuscript.
However, we agree that the list of references is overloaded with self-citation and it was revised. We hope that the article looks more appropriate for the publication with the provided changes.
Point 2: Another remark concerns the fact that the introduction to the article is very brief, and there is no mention of other methods for diagnosing physical processes during the laser processing of materials. The authors should compare their proposed approach with other methods and justify its advantages.
Response 2: Thank you for pointing it out. We do agree with your proposal. The introduction was revised. We added the necessary data to compare the proposed approach with other monitoring methods.
Point 3: Moderate editing of English language required.
Response 3: Thank you for pointing it out. English was revised by the native speaker.
Reviewer 2 Report
Comments and Suggestions for Authors
The paper presents the results of research of the process of laser impact on material surface with registration of acoustic emission signals and use of computed tomography. The authors established a relationship between laser processing parameters, amplitude of AE signals, volumes of the well, the removed material and the material adhered to the surface. The obtained dependences can be used for monitoring of technological processes of laser treatment. The article can be published considering the comments below.
Broad comment
The authors would be well advised to include an explanation of why they used a low-frequency vibration accelerometer to collect AE data instead of the ultrasonic sensors usually used in acoustic emission registration in the frequency range of 10 kHz - 1 MHz. Also, it is not clear how they recorded signals in a frequency range that exceeded the operating frequency of the accelerometer they used.
Specific comments
Page 3. The paper doesn't describe the equipment and software used by the authors to collect the AE data.
Page 3. Fig. 1, a: the numbers indicating parts of the experimental set-up mentioned in the figure caption are not visible
Author Response
Response to Reviewer 2 Comments
Dear reviewer,
Thank you so much for your kind evaluation of our work. We agree with all your proposals and comments and have modified the manuscript accordingly.
We hope the manuscript will be suitable for publishing in Sensors and attract many potential journal readers with your comments. The introduced corrections in the text of the manuscript are marked green.
Kind regards,
Authors.
Reviewer comments
Point 1: The authors would be well advised to include an explanation of why they used a low-frequency vibration accelerometer to collect AE data instead of the ultrasonic sensors usually used in acoustic emission registration in the frequency range of 10 kHz - 1 MHz. Also, it is not clear how they recorded signals in a frequency range that exceeded the operating frequency of the accelerometer they used.
Response 1: Thank you so much for your kind remark. If we consider the frequency response of the accelerometer, it is much broader than the operating range (Figure 1 in attachments).
The operating range is a linear range where the accelerometer conversion coefficient is constant. For this range, the accelerometer's passport indicates a conversion factor that allows us to obtain vibration measurement results in natural units (m/s2). However, the frequency response of the accelerometer is much broader than the linear range, extending into the region of low and high frequencies. It is also possible to evaluate vibrations there, but only in conventional units or mV. Measurements in a series of experiments should be conducted in the same frequency range to compare them with each other. Then, it is possible to see how many times the results of different measurements differ in the frequency range beyond the linear range. It can be noted that the largest conversion coefficient is in the resonance region, which makes it possible to observe vibrations of small amplitude.
The relevant explanation is provided in the manuscript text.
Point 2: Page 3. The paper doesn't describe the equipment and software used by the authors to collect the AE data.
Response 2: Thank you for pointing it out. We agree that it was not specified in the manuscript's text. The software used is original and developed by our research group under the supervision of Prof. Mikhail Kozochkin, who has worked in the field of industrial development of monitoring systems for machine tools since the 1980s. The standard PC was used for collecting data.
Point 3: Page 3. Fig. 1, a: the numbers indicating parts of the experimental set-up mentioned in the figure caption are not visible.
Response 3: Thank you for pointing it out. The figure was revised.
